# Research Progress on the Mechanism, Monitoring, and Prevention of Cardiac Injury Caused by Antineoplastic Drugs—Anthracyclines

**DOI:** 10.3390/biology13090689

**Published:** 2024-09-03

**Authors:** Yuanyuan Chen, Wenwen Yang, Xiaoshan Cui, Huiyu Zhang, Liang Li, Jianhua Fu, Hao Guo

**Affiliations:** 1Graduate School, China Academy of Chinese Medical Sciences, Beijing 100700, China; cyy960513@163.com (Y.C.); werweny@163.com (W.Y.); xscui33@163.com (X.C.); zhy17839980449@163.com (H.Z.); 18801386761@163.com (L.L.); 2Department of Cardiology, Shaanxi Academy of Traditional Chinese Medicine, Xi’an 710005, China

**Keywords:** anthracyclines, cardiotoxicity, mechanisms, monitoring, prevention, treatment

## Abstract

**Simple Summary:**

Anthracycline drugs, as a class of commonly used antitumor chemotherapeutics, play a significant role in the treatment of various cancers. However, they possess notable cardiotoxicity, which seriously undermines the quality of life for cancer patients. Therefore, early monitoring and mitigation of anthracycline-induced cardiotoxicity (AIC) have become urgent clinical issues to address. This study summarizes the mechanisms of AIC, encompassing oxidative stress, apoptosis, cardiac microenvironment remodeling, ferroptosis, and autophagy. Meanwhile, we have discovered that in addition to traditional medical imaging techniques that can monitor cardiotoxicity, emerging diagnostic tools such as advanced imaging technologies and human induced pluripotent stem cells demonstrate immense potential for early identification of subclinical toxicity and timely pharmacological intervention prior to the occurrence of irreversible cardiac damage. In terms of prevention and treatment of AIC, on the one hand, we can select appropriate drugs based on the molecular mechanisms of AIC to reduce cardiotoxicity. On the other hand, we can use derivatives with lesser cardiotoxicity, such as liposomes and nanoparticles, to prevent AIC, thereby improving the prognosis of cancer patients.

**Abstract:**

Anthracyclines represent a highly efficacious class of chemotherapeutic agents employed extensively in antitumor therapy. They are universally recognized for their potency in treating diverse malignancies, encompassing breast cancer, gastrointestinal tumors, and lymphomas. Nevertheless, the accumulation of anthracyclines within the body can lead to significant cardiac toxicity, adversely impacting both the survival rates and quality of life for tumor patients. This limitation somewhat restricts their clinical utilization. Determining how to monitor and mitigate their cardiotoxicity at an early stage has become an urgent clinical problem to be solved. Therefore, this paper reviews the mechanism of action, early monitoring, and strategies for the prevention of anthracycline-induced cardiotoxicity for clinical reference.

## 1. Introduction

Cancer stands as a primary cause of mortality worldwide, with its occurrence prevalence increasing year by year. Projections indicate that by 2070, the number of cancer patients worldwide may double compared to 2020, primarily due to demographic shifts, including population aging and growth [1]. Antitumor therapy has progressed rapidly in recent years with advances in medical technology. The age-standardized mortality rate of cancer has been slowly decreasing over the last decade, and the survival cycle of cancer patients is getting longer day by day [2]. The development of antineoplastic agents has notably improved the clinical survival outcomes of cancer patients, and anthracyclines (ANTs), ubiquitous chemotherapeutic agents, have demonstrated efficacy in treating various malignancies, including breast cancer, leukemia, and numerous other tumor types [3].

ANTs are a class of broad-spectrum antitumor drugs, including zorubicin, adriamycin (doxorubicin), epirubicin, pirubicin, and idarubicin. These drugs inhibit the growth and division of tumor cells by interfering with the replication and transcription process of DNA, thus achieving antitumor effects [4]. However, ANTs also have significant cardiotoxicity, and the cardiotoxicity they cause severely reduces the quality of survival of cancer patients. Long-term follow-up studies have found that anthracycline-induced cardiac damage has surpassed tumor recurrence as the main cause of death in cancer patients [5]. Data showed that up to 45.61% of pediatric patients applying anthracycline-based chemotherapy regimens developed cardiotoxicity [6].

Anthracycline-induced cardiotoxicity (AIC) is the result of a variety of molecular processes focusing on altered cardiomyocyte function and pathological cell death. AIC can be classified based on its onset time: acute, chronic, or delayed. Acute AIC typically arises within hours or days post-medication, often presenting with cardiac conduction abnormalities and arrhythmias; chronic AIC emerges within a year after drug administration, characterized by left ventricular dysfunction that may progress to heart failure; and delayed-onset AIC occurs several years after drug administration and manifests as heart failure, cardiomyopathy, arrhythmias, etc. [7]. The pathogenesis of AIC involves a variety of complicating factors. This article comprehensively reviews the advancements in AIC considering three aspects: pathogenesis, monitoring, and prevention, with a view to providing fresh insights and guidelines for the diagnosis and treatment of AIC.

## 2. Pathological Mechanisms of Cardiac Injury Due to Anthracyclines

### 2.1. Oxidative Stress

Oxidative stress arises from an imbalance between reactive oxygen species (ROS) and the body’s antioxidant defenses, constituting a pivotal mechanism underlying AIC [8]. ROS have a ‘double-edged sword’ effect in the organism, as a slight increase in ROS can induce a protective effect by triggering redox signaling, whereas excessive ROS production can deplete endogenous antioxidants and cause severe cardiomyocyte damage [9]. ANTs have a quinone structure that is mediated by other oxidoreductases, allowing the drug to function as an electron acceptor and trigger ROS generation. When ROS are produced in excess, the imbalance between their formation and elimination causes oxidative stress, which leads to lipid peroxidation, damages cell membranes, and ultimately causes cardiomyocyte death (Figure 1) [10].

#### 2.1.1. Mitochondria-Mediated ROS Generation

In the organism, mitochondria emerge as the primary target for ANTs [11]. The mitochondrial electron transport chain (ETC) is the main site of ROS production in mitochondria, accounting for about 90% or more of the total ROS in the organism [12]. ANTs accumulate in cardiomyocyte mitochondria after entering the organism. The anthraquinone structure contained in ANTs is reduced to semiquinone radicals under the action of various reductases and NADPH oxidases and then undergoes a series of electron transfer processes to generate ROS, including superoxide anion (O_2_^−^) and hydroxyl radical (OH^−^). An overabundance of these ROS within cardiomyocytes can instigate oxidative stress, which in turn triggers cardiomyocyte injury [13].

In addition, ANTs can inhibit respiratory chain function by forming complexes with high affinity for cardiolipin on the inner mitochondrial membrane. Mitochondrial damage in turn may lead to an increase in mitochondrial membrane permeability, prompting the release of Cytochrome c (Cyt c) from mitochondria, which mediates a rapid loss of mitochondrial membrane potential. This positive feedback regulatory mechanism further increases the production of ROS [14]. ROS have the ability to cause lipid peroxidation of mitochondrial membranes leading to disruption of membrane structure and dysfunction, which can further exacerbate the impairment of energy metabolism and create a vicious cycle. Additionally, ROS can also trigger mitochondrial DNA (mtDNA) mutation and damage, affecting the function and stability of mitochondria and thus exacerbating cardiomyocyte damage and apoptosis.

#### 2.1.2. NADPH Oxidase-Mediated ROS Generation

NADPH oxidase (NOX), a pivotal transmembrane oxidoreductase family, constitutes another crucial ROS-generating source. It facilitates the transfer of electrons from NADPH on one side of the membrane across the membrane to oxygen molecules on the other side, generating ROS. The human genome encodes seven NOX isoforms: NOX1-5, DUOX1, and DUOX2, with NOX2 emerging as the dominant player in AIC and NOX-induced ROS generation [15].

It was shown that after one week of doxorubicin (Dox) treatment, wild-type (WT) mice exhibited decreased survival, reduced cardiac function, a significant increase in NADPH-dependent superoxide formation, and increased myocardial oxidative stress, whereas in NOX2^−/−^ mice, these alterations were inhibited [16]. Another study revealed heightened NADPH oxidase activity along with upregulated NOX2 and NOX4 mRNA expression in Dox-treated mouse cardiac tissues. Notably, ROS originating exclusively from NOX2 were pinpointed as significant contributors to the progression of Dox-elicited cardiac damage and remodeling processes [17].

#### 2.1.3. NOS-Mediated ROS Generation

Nitric oxide (NO), a key mediator of anthracycline-induced oxidative stress [18], is enzymatically synthesized by nitric oxide synthase (NOS), which exists in three distinct isoforms: endothelial NOS (eNOS), inducible NOS (iNOS), and neuronal NOS (nNOS) [19]. In cardiomyocytes treated with ANTs, upregulation of both iNOS and eNOS levels has been observed. Firstly, ANTs can promote the generation of semiquinone radicals by binding to eNOS reductase, which reduces O_2_ to O_2_^−^. Then, the imbalance between generated O_2_^−^ and NO levels leads to ANTs binding to eNOS reductase, forcing eNOS to uncouple into monomers, which promotes an increase in O_2_^−^ and a decrease in NO production, leading to the generation of more ROS [20].

Neilan et al. [21] verified the pivotal role of eNOS in Dox-mediated oxidative stress through experiments involving eNOS knockout mice, which manifested low levels of ROS after Dox treatment. Their findings solidified eNOS as a crucial factor in the pathogenesis of left ventricular dysfunction following Dox exposure. More importantly, congenital eNOS deficiency enhanced the cardiotoxicity of the combination of adriamycin and trastuzumab in an acute mouse model of chemotherapy-induced cardiomyopathy [22]. In addition to eNOS, ANTs increased iNOS expression as well as eNOS activation and induced nitrooxidative stress, thereby increasing ROS levels in cardiac tissue [23].

#### 2.1.4. Iron-Mediated ROS Generation

In physiological states, intracellular iron is generally stored in iron storage proteins, maintaining a relatively low Fe^2+^ concentration and ensuring oxidative homeostasis through efflux via iron transport protein 1 [24]. However, the administration of ANTs leads to the release of a large quantity of iron ions in cardiomyocytes, and the accumulation of excess Fe^2+^ causes oxidative damage by generating excessive ROS directly via the Fenton reaction. Notably, ANTs possess a high affinity for iron, forming complexes that further elevate ROS levels [10]. Research has demonstrated that after receiving a Dox injection, mice exhibited marked cardiac dysfunction and increased production of ROS. Moreover, the administration of the iron inhibitor idebenone resulted in a significant reduction in iron levels and oxidative stress injury, indicating a critical role for iron in AIC and the possibility of iron reduction as a preventative therapeutic approach [25].

### 2.2. Apoptosis

Apoptosis, a genetically orchestrated process, represents a precise and self-contained mode of cell death essential for preserving internal environmental balance. It encompasses a cascade of gene activations, expressions, and regulatory mechanisms, culminating in an active and structured cell demise [26]. In the heart, the excessive occurrence of apoptosis disrupts the homeostasis of cardiomyocytes, leading to alterations in cardiac structure and function, which in turn leads to cardiac disease. Studies have shown that ANTs trigger two main apoptotic pathways—the mitochondrial (endogenous) apoptotic pathway and the death receptor (exogenous) apoptotic pathway (Figure 2) [27].

#### 2.2.1. Activation of the Mitochondrial Apoptotic Pathway

Mitochondria serve as central regulators in apoptosis, with a crucial step being the irreversible permeabilization of the mitochondrial outer membrane, which is intricately governed by the BCL2 family of proteins [28]. When mitochondria are stimulated by various anthracycline-induced stress signals (e.g., DNA damage, growth factor deficiency, oxidative stress, etc.), BAX is activated to undergo a structural transformation, causing a decrease in the mitochondrial membrane potential and prompting the opening of the permeability transition pore. This allows for the release of pro-apoptotic molecules, including Cyt c, into the cytoplasm, and Cyt c binds to apoptotic protease activator 1 (Apaf-1) to form an ATP-dependent apoptotic complex. Next, the precursor Caspase-9 is recruited to the complex and undergoes self-cleavage activation, which in turn triggers a cascade of caspases leading to apoptotic cell death [29].

#### 2.2.2. Activation of the Death Receptor Apoptotic Pathway

The exogenous pathway is initiated mainly through the binding of cell surface death receptors to extracellular death ligands [30]. These death receptors, which are members of the tumor necrosis factor receptor (TNFR) superfamily, feature a cysteine-rich extracellular region and an intracellular death domain (DD). When death receptors bind to specific death ligands, their intracellular portion undergoes a conformational change, recruiting junction proteins and precursor caspases to form a death-inducing signaling complex (DISC), which activates a cascade of caspases, leading to apoptosis [31]. Fas, belonging to the TNF family of receptors, is a pro-apoptotic death receptor. Dox initiates the external apoptotic pathway by stimulating nuclear factor-activated T cell-4 (NFAT4) and NF-κB, enhancing the expression of Fas/FasL and p53. Concurrently, it downregulates the expression of FLICE/caspase-8 inhibitory protein FLIP, thereby alleviating its inhibitory effect on Fas-mediated signaling [32].

### 2.3. Change in Cardiac Microenvironment

The cardiac microenvironment is the complex environment surrounding cardiomyocytes and their surrounding tissues, including a variety of components such as extracellular matrix (ECM), intercellular interactions, growth factors, cytokines, hormones, signaling molecules, etc. The cardiac microenvironment has an important impact on the value added, differentiation, and function of cardiomyocytes, as well as the overall structure and function of the heart [33]. When the cardiac microenvironment is altered, a series of pathophysiological processes may be triggered, leading to the occurrence and development of cardiac diseases.

ANTs activate cardiac fibroblasts, promote their transformation into myofibroblasts, and secrete large amounts of extracellular matrix proteins, which cause dynamic changes in the cardiac ECM, including ECM synthesis and degradation, leading to ECM remodeling [34]. An imbalance in the deposition and degradation of ECM proteins alters the mechanical properties of the tissue, making it stiff and non-compliant, further affecting cardiac function. During myocardial fibrosis, damaged myocytes and activated immune cells release large amounts of inflammatory response mediators such as cytokines, chemokines, and reactive oxygen radicals (Figure 3) [35]. Studies have shown that after ANT injury to the myocardium, nuclear factor is activated within 1 h, which continuously triggers the synthesis of a variety of inflammatory cytokines, such as inflammatory factors like TNF-α, IL-1β, and IL-6 [36,37]. These inflammatory cytokines subsequently coordinate the pro-inflammatory microenvironment and re-recruit leukocyte and macrophage infiltration, exacerbating myocardial inflammation and further exacerbating the process of myocardial injury and fibrosis, forming a vicious cycle. Recent studies have shown that the nucleotide-binding domain-like receptor protein 3 (NLRP3) inflammasome is associated with Dox-induced cardiotoxicity. Interestingly, decreasing NLRP3 inflammasome activity attenuated Dox-induced cardiotoxicity [38]. Another study showed that C1q/tumor necrosis factor-related protein 5 (CTRP5) overexpression protects the heart from Dox-induced oxidative stress and inflammatory injury by inhibiting TLR4/NLRP3 signaling [39].

### 2.4. Ferroptosis

Ferroptosis, a recently characterized mode of cell death, is fueled by iron-catalyzed lipid peroxidation. This process intertwines with iron, lipid, and glutathione metabolism [40]. ANTs promote ferroptosis in cardiomyocytes via dual mechanisms: on the one hand, they can modulate the antioxidant system to directly facilitate lipid peroxidation; on the other hand, iron ‘overload’ triggers the Fenton reaction, initiating the onset of ferroptosis (Figure 4) [41].

#### 2.4.1. Lipid Peroxidation

Lipid peroxidation, a process central to ferroptosis, involves the oxidation of unsaturated fatty acids abundant on cell membranes, facilitated by Fe^2+^ or lipoxygenases [42]. Typically, cystine enters cells via the cystine/glutamate reverse transporter (System Xc^−^), followed by reduction to cysteine in the glutathione (GSH) or thioredoxin reductase 1 (TXNRD1)-dependent cystine-reduction pathway, which facilitates the production of GSH. GSH is a potent reductant that acts as a glutathione peroxidase 4 (GPX4) cofactor that promotes the intracellular reduction of phospholipid hydroperoxides (PLOOHs) to the corresponding alcohols of PLOOHs (PLDHs). Lipid peroxides produced by cellular metabolism can then be reduced to non-toxic lipid alcohols, thus avoiding intracellular lipid peroxidation [41]. Li et al. corroborated ferroptosis as a contributor to Dox-induced chronic cardiotoxicity by RNA sequencing, and demonstrated through in vivo experiments in mice that Dox could reduce lipid peroxide scavenging by the System Xc^−^/GSH/GPX4 system by inhibiting the SLC7A11 subunit of System Xc^−^, which led to a decrease in GSH synthesis and thus promoted ferroptosis [43].

#### 2.4.2. Iron Accumulation

There are two oxidation states of iron in the body: Fe^2+^ and Fe^3+^. Fe^3+^ is transported into the cell from outside via the transferrin receptor TFR on the cell membrane. The intracellular PH environment is acidic, resulting in the freeing up of Fe^3+^ and its reduction to Fe^2+^ by the ferric reductase enzyme STEAP3. The Fenton reaction of the cellular Fe^2+^ with the peroxides generates Fe^3+^ and peroxyl radicals that attack lipid molecules and oxidize them to lipid peroxides [44]. Studies have shown that adriamycin is inserted into mtDNA, accumulates in mitochondria, and then downregulates 5′-aminolevulinic acid synthase 1 (Alas1) to disrupt hemoglobin synthesis, thereby impairing iron utilization and leading to mitochondrial iron overload in cardiomyocytes, demonstrating that Dox and iron accumulation in mitochondria synergistically induces ferroptosis in cardiomyocytes [45]. In addition, iron ions promote ROS production through mechanisms such as the Fenton reaction, which in turn further promotes lipid peroxidation and cell membrane disruption. Another study found that in Dox-treated cardiomyocytes, Dox was concentrated in mitochondria, leading to an increase in mitochondrial iron and ROS, and furthermore, treatment by upregulating the expression of mitochondrial iron-exporting proteins or chelating iron could reduce Dox-induced cardiotoxicity [46].

### 2.5. Autophagy

Autophagy is a catabolic process that provides energy and promotes material recycling and cellular self-renewal through lysosomal degradation of abnormally functioning or misfolded proteins and damaged organelles [47]. Autophagy exhibits a dual role in the onset and development of AIC. Physiological levels of autophagy ensure normal cardiomyocyte function and are considered a self-protective mechanism of the organism; however, excessive autophagy can cause damage to cardiomyocytes.

Some researchers have suggested that ANTs cause upregulation of cardiac autophagy leading to cardiotoxicity. On the one hand, ANTs activate Beclin-1 and promote autophagic vesicle formation [48]; on the other hand, Kobayashi et al. [49] demonstrated that ANTs significantly increased autophagic flux in cardiomyocytes and elevated protein levels of LC3-II. More importantly, AIC was exacerbated by the activation of autophagy but attenuated by the inhibition of autophagy, suggesting that autophagy contributes to AIC. However, other studies have reported that ANTs cause cardiotoxicity by inhibiting autophagy. It has been shown that ANTs inhibit the phosphorylation of AMPK and suppress autophagy by inhibiting autophagy initiation and suppressing autophagic flux processes [50]. Dox also blocks autophagic flux and inhibits autophagy initiation in cardiomyocytes by disrupting lysosomal acidification and lysosomal function. In addition, in vitro assays have shown that Dox upregulates the pro-autophagy protein Beclin-1 and induces apoptosis in cardiomyocytes [51]. The ambiguous role of autophagy in AIC may stem from various factors, such as the challenges in accurately assessing autophagic flux, the variability in Dox dosage and treatment duration, and the dynamic nature of the pathological process. Further research is necessary to elucidate the precise mechanisms and conditions under which autophagy exerts protective or detrimental effects in AIC.

### 2.6. Other

Topoisomerase II (Top II) is an enzyme that regulates DNA stranding and has two isoforms, of which TopII α is highly expressed in tumor tissues, whereas TopII β is highly expressed in myocardial tissues. The binding of ANTs to Top II α in tumor cells inhibits the entry of tumor cells into the G1/G2 phase, inhibits DNA replication, and leads to apoptosis, which is recognized as the molecular cornerstone of antitumor activity. In contrast, binding to Top II β can both activate the apoptotic signaling pathway, affecting cardiomyocyte oxidative phosphorylation and mitochondrial biosynthesis responses, and cause DNA double-strand cleavage, triggering cardiomyocyte death [52]. Experiments have shown alleviated Dox-induced heart failure in specific TopII β knockout mice, suggesting that Dox-induced cardiotoxicity is mediated by topoisomerase-II β in cardiomyocytes [53].

There is also a relationship between the occurrence of AIC and gender and age. Specifically, women and children are more likely to develop cardiotoxicity with ANTs than men and adults. Several studies have suggested that gender differences may depend on the lack of a protective effect of sex hormones against ANT-induced myocardial injury [54]. In addition, metabolic syndrome is a recognized risk factor for cardiovascular disease, which in turn is associated with an increased risk of cancer in adults. After the application of ANTs, oxidative stress and inflammatory responses may further induce cardiac injury in patients with metabolic syndrome [55]. Therefore, in the clinical application of ANTs, the patient’s age, gender, and disease should be taken into account.

## 3. Methods of Monitoring AIC

ANTs can cause cardiac damage and therefore cardiac monitoring is required when using these drugs to ensure patient safety. Below is a list of common monitoring methods (Table 1).

### 3.1. Electrocardiogram (ECG)

An electrocardiogram (ECG) monitors the electrical activity of the heart, and performing ECGs at regular intervals offers insights into the electrophysiological status of the heart before, during, and after ANT therapy. An ECG can detect any cardiotoxicity-induced abnormalities, including tachycardia at rest, alterations in ST-T segments, conduction disturbances, QT interval prolongation, and arrhythmias. Of these, the prolongation of the QT interval, which primarily represents the ventricular repolarization process, is of great significance. Under normal conditions, this value can vary somewhat in each lead, but the difference between the QT intervals in each lead reflects the synchrony of ventricular myocardial repolarization and potentiometric stability [56].

Research suggests that QT interval prolongation varies from 0% to 22% among ANT-treated patients, with severe prolongation (QTc exceeding 500 ms) observed in 0% to 5.2% of cases [57]. Studies have also demonstrated that cardiac insufficiency may occur early in the use of ANT chemotherapeutics, with low inter-patient variability in QT interval dispersion [58]. Therefore, QT interval dispersion may serve as an earlier predictor of AIC. However, these abnormalities have poor specificity and are confounded by many other factors. More importantly, these ECG changes are transient and do not detect chronic alterations in cardiac function.

### 3.2. Echocardiography

Echocardiography is a widely used non-invasive test for the detection of AIC. The new international consensus defines cardiotoxicity as a significant LVEF decline (>10%) resulting in an LVEF below 50%, and the results remain abnormal when reviewed 2–3 weeks after the earliest decrease [59]. Therefore, the occurrence of cardiotoxicity is often determined clinically based on a decrease in LVEF [60]. However, serial measurements of LVEF have important limitations, including physiological timing, operator-related variability, and hemodynamic load dependency, limiting early detection of subclinical myocardial damage.

Recent studies have justified that speckle tracking imaging (STI) is more reproducible than echocardiographic assessment and is more conducive to the assessment of progressive cardiotoxicity [61]. STI is a new technique developed in the field of ultrasound; it can continuously track the motion of the target myocardium and trace the spatial trajectory of speckles during the cardiac cycle, and it can assess LV systolic function by measuring global area strain (GAS), global longitudinal strain (GLS), global radial strain (GRS), and global circumferential strain (GCS) [62]. Among them, GLS is the most sensitive indicator of subclinical myocardial injury, and a decrease of more than 15% in GLS is often taken as an indicator of cardiotoxicity [63]. Meta-analyses have shown that GLS decreases earlier than LVEF, and more importantly, early changes in GLS can predict future deterioration in overall LV systolic function. However, its poor reproducibility and complexity limit its widespread use [64].

### 3.3. Biomarkers

Biomarkers are convenient, economical, and ideal for monitoring early AIC. Cardiac serum biomarkers commonly used in clinical practice include cardiac troponin (cTn), creatine kinase isoenzymes (CK-MB), and brain natriuretic peptide (BNP). Among them, cTn and BNP exhibit heightened sensitivity, enabling the early detection of subclinical cardiomyocyte alterations and a predictive assessment of left ventricular dysfunction subsequent to anthracycline therapy [65].

Cardiac troponin T (cTnT) and cardiac troponin I (cTnI) are exclusive structural proteins of the myocardium, serving as definitive indicators of myocardial damage. They are highly sensitive and specific for monitoring early AIC, and their peaks can reflect the extent of myocardial damage. BNP and NT-proBNP are secreted by the ventricular myocardium, and their levels vary according to ventricular wall tone, which has a negative feedback effect on ventricular filling. In heart failure, ventricular wall tension increases, and BNP and NT-proBNP secretion increase markedly, with the severity of the increase positively correlating with the severity of heart failure. Therefore, BNP and NT-proBNP are also important biomarkers for the assessment of cardiotoxicity. Cardinale et al. documented an increase in cTnI levels among 221 breast cancer patients receiving high-dose ANTs, underscoring its sensitivity and reliability as a myocardial injury marker with significant clinical and prognostic implications [66]. Similarly, Zardavas et al. [67] also observed that cTnI (>40 ng/L) was elevated in 13.6% of patients and cTnT (>14 ng/L) was elevated in 24.8%, both associated with a heightened risk of LVEF decline. Additionally, patients with notable LVEF reductions exhibited more pronounced NT-proBNP elevations from baseline.

### 3.4. Cardiac Magnetic Resonance Imaging (CMRI)

Cardiac magnetic resonance imaging (CMRI) stands as the gold standard for evaluating left ventricular volume and systolic function, playing a pivotal role in detecting cardiotoxicity due to chemotherapy and radiotherapy. It can detect myocardial damage even in the presence of a normal LVEF and can non-invasively provide information on tissue surface characteristics and valve function to determine the pathogenesis of myocardial dysfunction [68]. Therefore, CMRI allows early and accurate detection of AIC. Myocardial fibrosis is prevalent in patients with anthracycline-induced LV systolic dysfunction, and the measurement of T1 values can provide a quantitative index to evaluate the extent of diffuse myocardial fibrosis [69]. A comparison of pre- and post-T1 mapping revealed significantly elevated extracellular volume (ECV) scores among patients undergoing ANT therapy, with a robust correlation observed between ECV scores and the severity of cardiac insufficiency.

### 3.5. Radionuclide Ventriculography

Radionuclide ventriculography (RV) is a non-invasive imaging method that uses radionuclides to probe cardiac function and structure with good accuracy and high reproducibility for continuous quantitative assessment of LV systolic function, especially for continuous measurement of LVEF in cardiac disease models. It has been shown that RV correlates better with the degree of myocardial damage in the Dox-induced cardiotoxicity model compared to echocardiography. More importantly, only RV was independently associated with myocardial fibrosis in multiple regression analysis [70]. However, this imaging method is more expensive and carries some radiation risk, so there are some limitations in clinical application.

### 3.6. Other Monitoring Tools

High-risk patients for AIC can be effectively identified by utilizing advanced technologies such as genomics, proteomics, and other technologies. Several single-nucleotide polymorphisms (SNPs) linked to AIC have been discovered through candidate gene association studies and genome-wide association studies (GWASs). Specifically, the rs4673 polymorphism in the NADPH oxidase gene was found to confer protection against focal myocardial necrosis, while the rs1883112 polymorphism was strongly linked to cardiac fibrosis [71]. In addition, the SNP rs28714259 has been implicated in heightened vulnerability to anthracycline-induced heart failure, and this association was confirmed in a breast cancer trial setting [72]. Recent studies have also demonstrated the potential of utilizing patient-specific human induced pluripotent stem-cell-derived cardiomyocytes (hiPSC-CMs) as a predictive tool for assessing AIC risk [73]. The optimization of their isolation and culture techniques is expected to be used for the identification of genetic biomarkers and validation of SNPs. High-throughput screening in hiPSC-CMs may identify genetic modification strategies for cardioprotection and accelerate the development of new therapies.

## 4. Advances in the Treatment of Anthracycline-Induced Cardiotoxicity

Primary prevention strategies for AIC focus on minimizing the cardiac toxicity of the drug itself or employing derivatives that exhibit reduced cardiotoxic effects (Figure 5).

### 4.1. Pharmacological Treatments

#### 4.1.1. Inhibiting Oxidative Stress

Folic acid, a regulator of eNOS, has shown potential in mitigating Dox-induced cardiotoxicity. Research has demonstrated that continuous Dox injection over 10 days resulted in a 70% mortality rate in mice, which was reduced to 45% with folic acid pretreatment. The protective mechanism involves lowering the eNOS monomer-to-dimer ratio, preventing the increase in superoxide anions, reducing SOD, eNOS phosphorylation, and NO production. This restoration of eNOS coupling effectively reduces oxidative stress and subsequent myocardial damage induced by Dox [74]. Naringin, found in citrus fruits, has significant cardioprotective effects against Dox-induced cardiomyopathy. Research indicates that naringin enhances the activity of antioxidant enzymes such as GSH, SOD, and CAT in the hearts of Dox-treated rats. It also improves mitochondrial respiratory function by boosting the activity of mitochondrial complexes I, II, III, and IV, thereby reducing reactive oxygen species (ROS) production [75]. Curcumin, a polyphenolic compound, protects myocardial cells from Dox-induced damage by modulating mitochondrial permeability transition pores (mPTPs) and inhibiting ROS release [76].

#### 4.1.2. Inhibiting Apoptosis

Isoquercitrin, a natural flavonoid, possesses antioxidant and anti-apoptotic properties. It protects cardiomyocytes through modulation of the Phlpp1/AKT/Bcl-2 signaling pathway, reducing mitochondrial membrane permeability, inhibiting the release of Cyt c from the mitochondrial inner membrane to the cytoplasm, and preventing the formation of apoptotic bodies. This significantly reduces pirarubicin-induced cardiotoxicity [77]. Bitter almond aldehyde ethanol extract, made by soaking dried bitter almond kernel oil leaves in ethanol, protects against AIC by significantly reducing the expression of TGF, Cyt c, and apoptotic factors in P600 and P800 [78]. Saleh et al. [79] demonstrated that catechins reversed the Dox-induced decrease in myocardial antioxidant enzyme levels and reduced apoptosis in the mitochondrial pathway. This was further corroborated by Ozyurt et al., who demonstrated the ability of catechins to reduce lipid peroxidation and prevent Dox-induced cardiotoxicity in rats [80].

#### 4.1.3. Regulating the Cardiac Microenvironment

Changes in the cardiac microenvironment significantly contribute to AIC, with extracellular matrix (ECM) remodeling and inflammation playing crucial roles. Doxycycline, a versatile agent that possesses both anti-inflammatory properties and inhibits matrix metalloproteinases (MMPs), can prevent ECM degradation and fibrosis, alter myocardial energy metabolism, and mitigate Dox-induced cardiotoxicity [81]. Nerolidol has also demonstrated protective abilities against the development of chronic cardiotoxicity in rats exposed to Dox. Compared to the Dox group, nerolidol effectively reverses the increase in myocardial marker enzymes and prevents Dox-mediated oxidative damage, myocardial fibrosis, and inflammation. This protection is achieved through the activation of the PI3K/Akt/Nrf2 pathway and inhibition of the NF-κB/MAPK pathway [82]. A study by Sumeet Kumar et al. [83] demonstrated that betaine, by activating AMPK and Nrf2 and inhibiting TGF-β expression and activity, reduces oxidative stress, inflammation, and fibrotic responses, to prevent cardiac damage. Additionally, matrine has been found to regulate the RPS5/p38 signaling pathway, improving pathological cardiac fibrosis and cardiac dysfunction in mice [84].

#### 4.1.4. Inhibiting Ferroptosis

Dexrazoxane, an iron-chelating agent, effectively counteracts iron-induced oxidative stress and holds the distinction of being the first FDA-approved medication solely dedicated to addressing AIC. It protects the heart by chelating free iron, thereby inhibiting the formation of ROS. Despite its effectiveness in reducing cardiotoxicity, dexrazoxane’s clinical application remains controversial due to concerns about its efficacy, safety, potential impact on chemotherapy effectiveness, suitability for various patient populations, and cost-effectiveness. While it has been shown to significantly reduce the incidence of cardiotoxicity in children, some studies suggest a potential increased risk of secondary cancers in pediatric patients using dexrazoxane [85]. Research by Chen et al. [86] indicates that salidroside, a compound derived from Rhodiola rosea, activates AMPK-dependent signaling pathways, limits iron accumulation, and restores GPX4-dependent antioxidant capacity. This modulation of ferroptosis can alleviate cardiac dysfunction, reduce ferroptosis, and prevent fibrosis, making salidroside a potential novel therapeutic agent for managing Dox-induced cardiotoxicity.

#### 4.1.5. Modulating Autophagy

SAR405, a novel autophagy inhibitor, has been shown to reduce Dox-induced cytotoxicity when used in combination with dexamethasone [87]. Chloroquine offers cardioprotection by inhibiting the fusion of autophagosomes with lysosomes, thereby preventing excessive autophagic degradation [88]. These results suggest that inhibiting autophagy can mitigate the cardiotoxic effects of ANTs. Conversely, other studies indicate that ANTs promote autophagy. For instance, rapamycin, an mTOR inhibitor that enhances autophagy, has demonstrated protective effects against Dox-induced cardiotoxicity [89]. Similarly, low-dose colchicine, which promotes autolysosomal degradation, reduces Dox-induced cardiotoxicity by regulating microtubules [90]. Metformin, a common diabetes medication, restores PRKAA/AMPK phosphorylation and reactivates autophagic flux in cardiomyocytes, offering protection from damage and reducing cardiotoxicity [91]. Additionally, dihydroartemisinin promotes autophagy and minimizes lysosomal damage, helping maintain cellular homeostasis and preventing Dox-induced injury [92]. These findings highlight the dual role of autophagy modulation—both inhibition and promotion—in reducing the cardiotoxicity associated with ANTs.

### 4.2. Novel Drug Delivery Systems

Recent advancements in drug delivery technologies have introduced various innovative methods such as liposomes, nanoparticles, exosomes, micelles, dendrimers, and microrobots. These technologies aim to enhance drug bioavailability and therapeutic indices while ensuring high safety. However, the biopharmaceutical aspects of these delivery systems are still not fully understood, presenting challenges in translating preclinical findings into clinical applications.

Liposomal Dox has been shown to minimize cardiac damage by inducing interferon-related DNA damage resistance, thus activating protective cellular pathways while maintaining the drug’s anticancer efficacy. This method offers a promising strategy for enhancing Dox’s safety in clinical settings [93]. Two clinically approved formulations of liposomal Dox exist: pegylated liposomal Dox (PLD, commercially known as Doxil/Caelyx or Lipdox), and non-pegylated liposomal Dox (NPLD, marketed as Myocet), each offering distinct advantages in therapeutic applications [94]. Using nanoparticles or mesenchymal stem-cell-derived exosomes as drug carriers can enhance drug delivery specificity and effectiveness, reducing cardiotoxicity. Yang et al. developed Dox dimeric prodrug nanoassemblies stabilized by trisulfide bonds, and they exhibited high drug loading, increased stability, and selective tumor targeting. These nanoassemblies facilitate drug release in the tumor microenvironment, improving efficacy while minimizing systemic toxicity [95]. Studies have shown that exosomes carrying miRNA-499a-5p can target CD38, a protein associated with cardiac stress, thus alleviating anthraquinone-induced cardiotoxicity [96]. 

Both polymeric micelles and dendrimers enhance drug solubility, stability, and targeted delivery. Light-responsive Poly-Dox-M polymeric micelles release Dox under specific wavelengths of light, improving drug delivery efficiency and reducing side effects [97]. Researchers have also developed red-light-responsive metallopolymer nanocarriers that combine conjugated and encapsulated drugs for precise control over drug release and enhanced efficacy [98]. Dendrimers, with their highly branched structure, improve Dox solubility, drug loading, stability, and controlled release, enhancing therapeutic outcomes and reducing side effects [99,100]. *Escherichia coli* Nissle 1917 can be used as a power source for microrobots due to its unique biological properties, enabling precise localization and penetration of tumor tissues, thereby improving drug delivery accuracy and therapeutic efficacy [101]. These technologies provide valuable references for developing new anthracycline derivatives with minimal or no cardiotoxicity, advancing the field of cancer therapy.

### 4.3. Non-Pharmacological Interventions

It has been shown that appropriate exercise can mitigate the heart damage caused by anthracycline chemotherapy [102,103]. A review of the literature suggests that exercise improves cardiac dysfunction in breast cancer patients receiving anthracycline-based chemotherapy, with cellular and molecular mechanisms including a reduction in oxidative stress, the protection of mitochondria, and the modulation of cardiac and vascular adaptations [104]. Recent studies have shown [105] that in Dox-impaired hearts, exercise promotes the differentiation of myeloid cells into pericytes and endothelial cells, elevates systolic and diastolic blood flow, and protects cardiac function. These findings suggest that the increased number of endothelial cells after exercise may contribute to the maintenance of open lumen and repair of cardiac tissue, revealing a mechanism by which exercise protects the cardiovascular system and ultimately alleviates Dox-induced cardiotoxicity.

## 5. Conclusions and Perspectives

The incidence of cardiovascular complications stemming from cancer treatment is projected to escalate steadily over the next few years, heralding the emergence of a potential cardio-oncology epidemic. ANTs, a class of commonly used antitumor chemotherapeutic agents, play an important role in the treatment of many cancers. However, AIC has been a major clinical concern. Despite our initial understanding of the molecular mechanisms behind AIC, progress in its diagnosis and treatment is still insufficient.

In this paper, we systematically summarize the mechanism of action of AIC, covering multiple levels of oxidative stress, apoptosis, cardiac microenvironmental remodeling, ferroptosis, and autophagy, with a view to providing more ideas for the prevention and treatment of AIC. AIC tends to be dose-dependent, progressive, and irreversible; therefore, early monitoring and aggressive treatment of cardiotoxicity induced by ANTs are of great significance. In recent years, in addition to the traditional medical impact techniques for the detection of cardiotoxicity, the application of emerging diagnostic tools such as advanced imaging techniques and human induced pluripotent stem cells has demonstrated the great potential for the early identification of subclinical toxicity and the implementation of timely pharmacological interventions before the onset of irreversible cardiac damage. In the prevention and treatment of AIC, on the one hand, we can reduce AIC by selecting appropriate drugs according to the molecular mechanism of AIC. On the other hand, less cardiotoxic derivatives such as liposomes, nanoparticles, micelles, and dendritic polymers can be used to prevent AIC.

It is crucial for the development of the field of oncology cardiology to effectively manage the interaction between the cardiovascular system and tumors, as well as to keep cardiotoxicity to a minimum during treatment. Therefore, in-depth studies on the complex network of molecular mechanisms of AIC are still ongoing, aiming at identifying new molecular targets, thus providing more effective drug choices for the treatment of various forms of cardiotoxicity and improving the prognosis of tumor patients.

## Figures and Tables

**Figure 1 biology-13-00689-f001:**
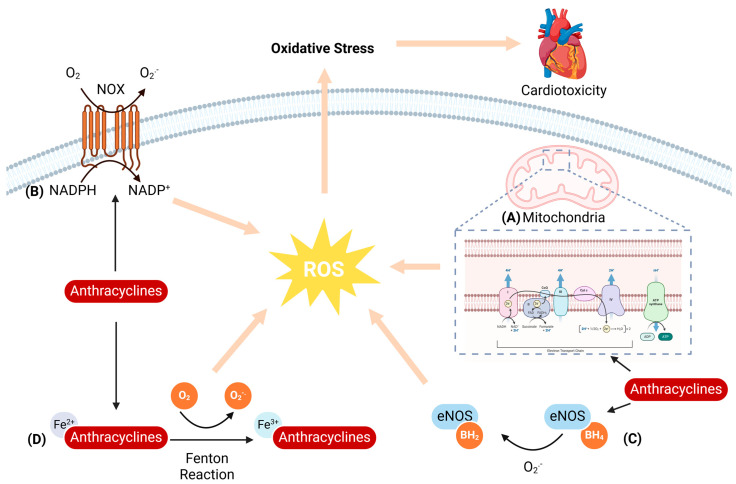
Schematic representation of anthracycline-induced oxidative stress. (**A**) Anthracyclines disrupt the mitochondrial electron transport chain, enhancing ROS generation. (**B**) Anthracyclines increase NADPH oxidase activity, accelerating electron transfer from NADPH to the plasma membrane and reducing O_2_ to form O_2_^.−^. (**C**) Anthracyclines uncouple eNOS, which in turn leads to decreased NO bioavailability and increased ROS production. (**D**) Anthracyclines cause intracellular iron overload in cardiomyocytes, and Fe^2+^ accumulation generates excessive ROS via the Fenton reaction.

**Figure 2 biology-13-00689-f002:**
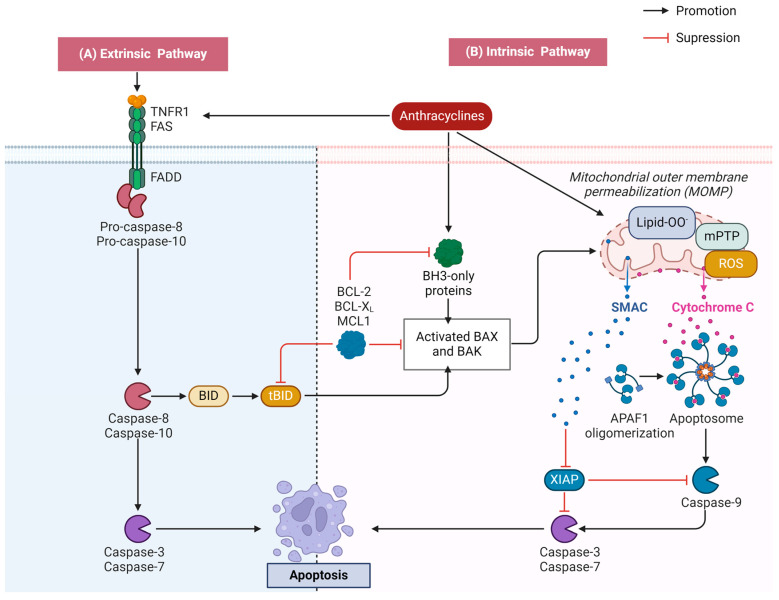
Schematic diagram of anthracycline-induced apoptosis. (**A**) Anthracyclines activate the death receptor and form the death-inducing signaling complex. This complex subsequently activates the caspase cascade reaction, ultimately resulting in exogenous apoptosis. (**B**) BAX is activated to undergo a conformational change when anthracyclines stimulate the mitochondria, which leads to a decline in the mitochondrial membrane potential and the opening of the permeability transition pore. This event facilitates membrane lipid peroxidation and prompts the release of pro-apoptotic factors, notably Cyt c, which in turn triggers the caspase cascade reaction, leading to endogenous apoptosis.

**Figure 3 biology-13-00689-f003:**
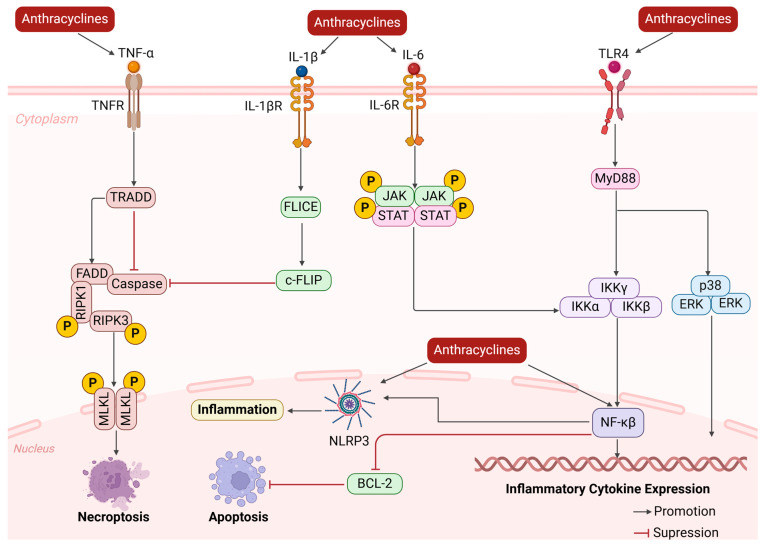
Schematic diagram of anthracycline-induced changes in the cardiac microenvironment. Anthracyclines lead to the release of large amounts of inflammatory mediators, exacerbating the inflammatory response, contributing to the remodeling of the ECM, causing apoptosis or necrosis of cardiomyocytes, and further exacerbating the process of myocardial injury and fibrosis.

**Figure 4 biology-13-00689-f004:**
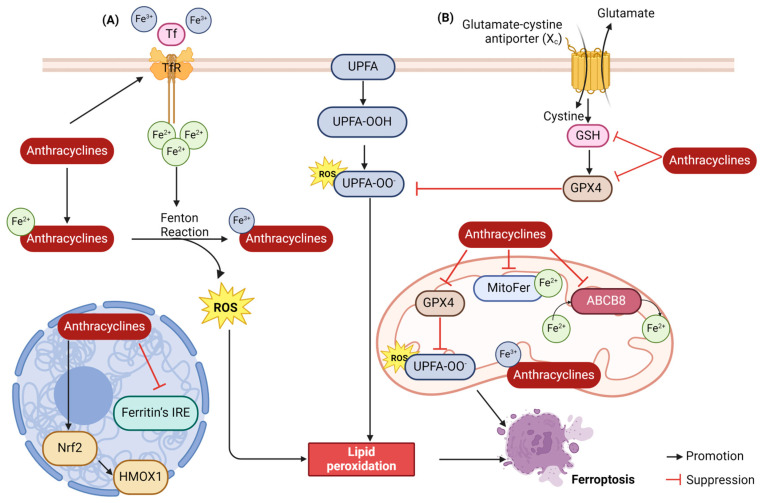
Schematic representation of anthracycline-induced ferroptosis. (**A**) Iron “overload” triggers the Fenton reaction, initiating ferroptosis. (**B**) Anthracycline-triggered alterations in the antioxidant system levels contribute to lipid peroxidation, which subsequently initiates the process of ferroptosis.

**Figure 5 biology-13-00689-f005:**
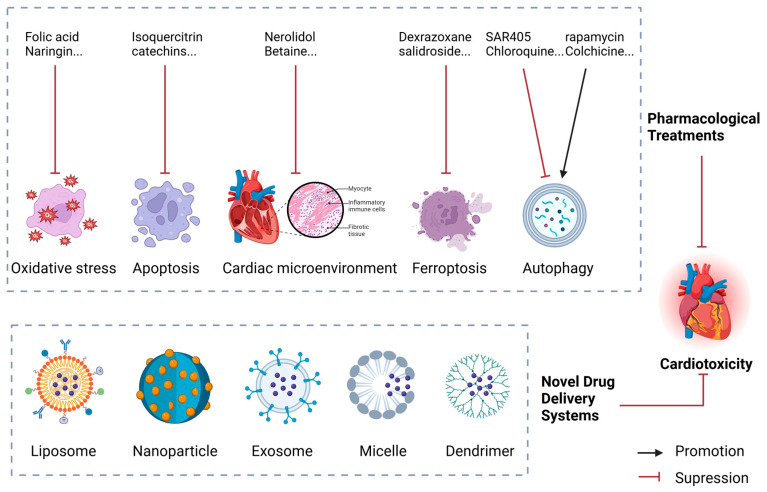
Schematic diagram of the prevention and treatment strategy for anthracycline-induced cardiotoxicity.

**Table 1 biology-13-00689-t001:** Methods for monitoring anthracycline-induced cardiotoxicity.

Methods	Diagnostic Criteria	Superiority	Inferiority
ECG [56,57,58]	1. ST segment change 2. QT prolongation	Accessible and economical	Insufficient specificity
Echocardiography	1. A decrease in LVEF of >10% and LVEF < 50% [59,60] 2. A decrease of GLS > 15% [61,62,63,64]	Provides information about structure and function	Unable to identify subclinical cardiomyocyte structural alterations
Biomarkers [65,66,67]	1. Continuous cTnI elevation 2. BNP elevation	Sensitive and repeatable	Insufficient evidence
CMRI [68,69]	A decrease in LVEF of >10% and LVEF < 50%	1. Presents comprehensive information encompassing both the structure and function 2. Excellent repeatability	Expensive and stringent adherence prerequisites
RV [70]	A decrease in LVEF of >10% and LVEF < 50%	Surpasses standard two-dimensional echocardiography in terms of precision and reproducibility when measuring LVEF	Radiation exposure

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
