# Peer review of "Research Progress on the Mechanism, Monitoring, and Prevention of Cardiac Injury Caused by Antineoplastic Drugs—Anthracyclines"

_biology, 2024, doi:10.3390/biology13090689_

Round 1

Reviewer 1 Report

Comments and Suggestions for Authors

In this review the authors review the literature on an important aspect of cardio-oncology. Anthracyclines are drugs of strategic importance in anti-tumor therapy, despite presenting serious problems in the cardiovascular field. The authors investigate this aspect in detail on multiple levels.

points:

1- the authors indicate a series of pro-inflammatory cytokines, it would be appropriate to indicate the involvement of Nlrp3 as reported in the literature

2- among the important problems in the oncology field there is the presence of metabolic problems and sex-related differences, it would be appropriate for the authors to consider these aspects in relation to the role of anthracyclines (PMID: 32708201, PMID: 31256318)

Author Response

Reviewer 1

Comments 1: the authors indicate a series of pro-inflammatory cytokines, it would be appropriate to indicate the involvement of Nlrp3 as reported in the literature

Response 1: Thanks for your suggestion. We added a short explanation of NLPR3 “Recent studies have shown that the nucleotide-binding domain-like receptor protein 3 (NLRP3) inflammasome are associated with Dox-induced cardiotoxicity. Interestingly, decreasing NLRP3 inflammasome activity attenuated Dox-induced cardiotoxicity[1]. Another study showed that C1q/tumor necrosis factor-related protein 5 (CTRP5) overexpression protects the heart from Dox-induced oxidative stress and inflammatory injury by inhibiting TLR4/NLRP3 signalling[2]” in lines 212-217. We believe that the comprehensiveness of our study has been greatly improved following your suggestions.

[1] Maayah, Z.; Takahara, S.; Dyck, J., The beneficial effects of reducing NLRP3 inflammasome activation in the cardiotoxicity and the anti-cancer effects of doxorubicin. Arch Toxicol 2021,95(1),1-9.https://dx.doi.org/10.1007/s00204-020-02876-2

[2] Zhang, Z.; Peng, J.; Hu, Y.; Zeng, G.; Du, W.; Shen, C.,CTRP5 Attenuates Doxorubicin-Induced Cardiotoxicity Via Inhibiting TLR4/NLRP3 Signaling. Cardiovasc Drugs Ther 2023,31.https://dx.doi.org/10.1007/s10557-023-07464-x

Comments 2: Among the important problems in the oncology field there is the presence of metabolic problems and sex-related differences, it would be appropriate for the authors to consider these aspects in relation to the role of anthracyclines (PMID: 32708201, PMID: 31256318)

Response 2: Thanks for your suggestion. We apologize for not considering the effects of gender and metabolism on anthracycline-induced cardiotoxicity. According to your sunggestion,we have added a short explanation of the effects of metabolic problems and sex-related differences on anthracycline cardiotoxicity in lines 306-315: “There is also a relationship between the occurrence of AIC and gender and age. Specifically, women and children are more likely to develop cardiotoxicity with ANT than men and adults. Several studies have suggested that gender differences may depend on the lack of protective effect of sex hormones against ANT-induced myocardial injury [1]. In addition, the metabolic syndrome is a recognised risk factor for cardiovascular disease, which in turn is associated with an increased risk of cancer in adults. After the application of ANT, oxidative stress and inflammatory responses may further induce cardiac injury in patients with metabolic syndrome[2]. Therefore, in the clinical application of ANT, the patient's age, gender and their own disease should be taken into account.” Thanks for your suggestions to improve the quality of the study.

[1] Dessalvi, C.; Pepe, A.;Penna, C.;Gimelli, A.,Sex differences in anthracycline-induced cardiotoxicity: the benefits of estrogens. Heart Fail Rev 2019,24(6),915-925.https://dx.doi.org/10.1007/s10741-019-09820-2

[2] Mercurio, V.; Cuomo, A.;Dessalvi, C.; Deidda,M.,Redox Imbalances in Ageing and Metabolic Alterations: Implications in Cancer and Cardiac Diseases. An Overview from the Working Group of Cardiotoxicity and Cardioprotection of the Italian Society of Cardiology (SIC). Antioxidants (Basel) 2020,9(7),641.https://dx.doi.org/10.3390/antiox9070641

Reviewer 2 Report

Comments and Suggestions for Authors

Chen Y et al demonstrated the existing knowledge on mechanisms of action, early monitoring and prevention strategies of anthracycline-induced cardio-toxicity. In general, it is a well-organized review providing interesting information. However, some minor changes or additions are necessary.

Specifically, it would be very interesting to add the potential mechanisms of exercise on endothelial progenitor cells in HF, as part of the non-pharmacological interventions. There is recent and detailed literature regarding this issue.

Moreover, it would be really useful to change the format of the discussion part and separate it into limitations, future perspectives and conclusion so that to make it more clear.

Finally, an illustration including methods of monitoring and treatment strategies would increase the quality of the article.

Comments on the Quality of English Language

Minor english editing is required.

Author Response

Reviewer 2

Comments 1: It would be very interesting to add the potential mechanisms of exercise on endothelial progenitor cells in HF, as part of the non-pharmacological interventions. There is recent and detailed literature regarding this issue.

Response 1: Thanks for your suggestion. We added a short explanation of the potential mechanisms of exercise on anthracycline-induced cardiotoxicity and to cite the novel idea that exercise protects against Dox-induced cardiotoxicity by increasing the endothelial cells that promote vascular repair in lines 547-558: “ It has been shown that appropriate exercise can mitigate the heart damage caused by anthracycline chemotherapy[1, 2]. A review of the literature suggests that exercise improves cardiac dysfunction in breast cancer patients receiving anthracycline-based chemotherapy, with cellular and molecular mechanisms including reduction of oxidative stress, protection of mitochondria, and modulation of cardiac and vascular adaptations[3]. Recent studies have shown[4] that in Dox-impaired hearts, exercise promotes the differentiation of myeloid cells into pericytes and endothelial cells, elevates systolic and diastolic blood flow, and protects cardiac function. These findings suggest that the increased number of endothelial cells after exercise may contribute to the maintenance of open lumen and repair of cardiac tissue, revealing a mechanism by which exercise protects the cardiovascular system and ultimately Dox-induced cardiotoxicity.” Thank you for your input to make our article more meaningful!

[1] Willeke R, N.;  David, B.;  Martijn M, S.;  Neil K, A.;  Arco J, T.;  Wim H, v. H.;  Wim G, G.; Anne M, M., Efficacy of Physical Exercise to Offset Anthracycline-Induced Cardiotoxicity: A Systematic Review and Meta-Analysis of Clinical and Preclinical Studies. Journal of the American Heart Association 2021, 10 (17).http://dx.doi.org/10.1161/jaha.121.021580

[2] Lee, Y.;  Kwon, I.;  Jang, Y.;  Cosio-Lima, L.; Barrington, P., Endurance Exercise Attenuates Doxorubicin-induced Cardiotoxicity. Medicine and science in sports and exercise 2020, 52 (1), 25-36.http://dx.doi.org/10.1249/mss.0000000000002094

[3] Dozic, S.;  Howden, E. J.;  Bell, J. R.;  Mellor, K. M.;  Delbridge, L. M. D.; Weeks, K. L., Cellular Mechanisms Mediating Exercise-Induced Protection against Cardiotoxic Anthracycline Cancer Therapy. Cells 2023, 12 (9).http://dx.doi.org/10.3390/cells12091312

[4] Tao, R.; Kobayashi, M.; Yang, Y.; Kleinerman, E.,Exercise Inhibits Doxorubicin-Induced Damage to Cardiac Vessels and Activation of Hippo/YAP-Mediated Apoptosis. Cancers (Basel) 2021,13(11),2740.https://dx.doi.org/10.3390/cancers13112740

Comments 2: It would be really useful to change the format of the discussion part and separate it into limitations, future perspectives and conclusion so that to make it more clear.

Response 2: Thanks for your suggestion. We have modified conclusion from three aspects: limitations, conclusions and perspectives in lines 560-583.

Conclusions and perspectives

The incidence of cardiovascular complications stemming from cancer treatment is projected to escalate steadily over the next few years, heralding the emergence of a potential cardio-oncology epidemic. ANT, a class of commonly used antitumour chemotherapeutic agents, play an important role in the treatment of many cancers. However, AIC has been a major clinical concern. Despite our initial understanding of the molecular mechanisms behind AIC, progress in its diagnosis and treatment is still insufficient.

In this paper, we systematically summarize the mechanism of action of AIC, covering multiple levels of oxidative stress, apoptosis, cardiac microenvironmental remodeling, ferroptosis and autophagy, with a view to providing more ideas for the prevention and treatment of AIC. AIC tends to be dose-dependent, progressive and irreversible, therefore, early monitoring and aggressive treatment of cardiotoxicity induced by ANT are of great significance. In recent years, in addition to the traditional medical impact techniques for the detection of cardiotoxicity, the application of emerging diagnostic tools such as advanced imaging techniques and human induced pluripotent stem cells has demonstrated the great potential for the early identification of subclinical toxicity and the implementation of timely pharmacological interventions before the onset of irreversible cardiac damage. In the prevention and treatment of AIC, on the one hand, we can reduce AIC by selecting appropriate drugs according to the molecular mechanism of AIC. On the other hand, less cardiotoxic derivatives such as liposomes, nanoparticles, micelles and dendritic polymers can be used to prevent AIC.

It is crucial for the development of the field of oncology cardiology to effectively manage the interaction between the cardiovascular system and tumours, as well as to keep cardiotoxicity to a minimum during treatment. Therefore, in-depth studies on the complex network of molecular mechanisms of AIC are still ongoing, aiming at identifying new molecular targets, thus providing more effective drug choices for the treatment of various forms of cardiotoxicity and improving the prognosis of tumour patients.

Comments 3: An illustration including methods of monitoring and treatment strategies would increase the quality of the article.

Response 3: Thanks for your suggestion. We have added relevant illustrations. The methods for monitoring anthracycline-induced cardiotoxicity are summarised in Table 1 and the prevention and treatment strategy for anthracycline-induced cardiotoxicity are exhibited via Figure 5.

Comments 4: Comments on the Quality of English Language

Minor english editing is required.

Response 4: Thanks for your suggestion. We have tried our best to improved the quality of English Language.

Comments from Editor:

Comments 1:The authors are recommended to emphasise new progresses in this research field by comparing a similar prior publication: https://www.ncbi.nlm.nih.gov/pmc/articles/PMC10762801/

Response 1:This study addresses three aspects of anthracycline-induced cardiotoxicity: molecular mechanisms, monitoring methods, and therapeutic strategies. In contrast to similar previous publications, this paper, in addition to summarizing the role of traditional medical impact techniques for monitoring cardiotoxicity, highlights the great potential of emerging diagnostic tools such as advanced imaging techniques and the application of human induced pluripotent stem cells for the early identification of subclinical toxicity and implementation of timely pharmacological interventions prior to the onset of irreversible cardiac damage. On the other hand, this study aims to classify the medications that lessen anthracycline-induced cardiotoxicity based on the molecular mechanism of AIC. This classification approach will enable precise mechanism-target-drug therapy. Furthermore, the utilization of less cardiotoxic derivatives like liposomes and nanoparticles or non-pharmacological therapy like exercise to avoid anthracycline-induced cardiotoxicity is also described, which provides more ideas and strategies to prevent AIC.